# Exergames in Older Adult Community Centers and Nursing Homes to Improve Balance and Minimize the Risk of Falls in Older Adults: A Systematic Review and Meta-Analysis

**DOI:** 10.3390/healthcare11131872

**Published:** 2023-06-28

**Authors:** José Carlos Leal, Vinícius Silva Belo, Ingrid Morselli Santos, Rodrigo Vinícius Ferreira, Saulo Nascimento de Melo, Eduardo Sérgio da Silva

**Affiliations:** 1Graduate Program in Health Sciences, Federal University of São João Del Rei, CCO Campus, Divinópolis 35501-296, Minas Gerais, Brazil; leal.educacaofisica@gmail.com (J.C.L.); rpersonal17@hotmail.com (R.V.F.); saulomelobio@hotmail.com (S.N.d.M.); silvaedu@ufsj.edu.br (E.S.d.S.); 2Physical Education Course, University Center of Formiga, Formiga 35574-530, Minas Gerais, Brazil; 3Medicine Course, Federal University of São João Del Rei, CCO Campus, Divinópolis 35501-296, Minas Gerais, Brazil; ingridmorselli@yahoo.com.br

**Keywords:** older population, physical exercise, exergame, health benefits, gait function

## Abstract

There is a substantial gap in our knowledge regarding the efficacy of exergames on the reduction of fall risk in older adults. This systematic review analyzes the findings of clinical trials describing the efficacy of exergames to improve balance or reduce the risk of falls in individuals above 60 years of age who are residents in community centers or nursing homes. We searched Google Scholar, PubMed, and Embase up to January 2023. Initially, 52,294 records were screened. After applying the inclusion and exclusion criteria, 20 studies were included in this systematic review. Meta-analyses revealed statistically significant reductions in the risk of falls and improvements in balance. Exergaming tended to produce positive benefits according to the results obtained using different instruments (TUG, PPA, BBS, and others), control groups, and times of intervention. Nevertheless, a substantial proportion of studies exhibited a high risk of bias and only one had a long follow-up period. Although a large body of evidence supports the view that exergaming is suitable for reducing fall risk and improving balance in older adults, some gaps remain in our knowledge about such benefits.

## 1. Introduction

The proportion of older adults within the population is growing in both developed and developing countries as a result of a decline in the rates of mortality and fertility together with an increase in life expectancy [1,2]. In 2017, individuals aged 60 years or more accounted for 12.3% of the world’s population, but this percentage is expected to increase to 21.3% by 2050 [3]. In this regard, it is imperative to understand and identify the priority problems of the older population in order to establish new policies and actions to ensure their security and to provide the health care that they need [4].

Accidental falls are the major causes of disability, morbidity, mortality, and institutionalization among older adult citizens, resulting in substantial social impacts especially in countries with a large aging population [5,6]. Recent research has shown that one in three individuals aged more than 65 years has at least one fall per year [7], the causes of which are associated with intrinsic (mobility, muscle strength, clinical, and cognitive conditions) or extrinsic (physical, social, and behavioral) factors [8].

In addition to the harm caused to physical health, falls in older adults are of particular concern as they engender changes in family structure and dynamics due to the need for specialized care, hospitalization, or institutionalization, all of which generate enormous pressure on health services [5,9].

In healthcare settings, the most significant impacts of falls are an increased demand for emergency care, ambulance services, medical consultations, and diagnostic tests. Additionally, treating fall-related injuries incurs high costs, including hospitalization expenses, surgery, medication, and rehabilitation. Within the family structure, falls generate additional costs both in terms of medical assistance and care for the elderly individual [5]. The responsibility of caring for older adults often falls on the family, which requires significant time, energy, and financial resources. Depending on the severity of their injuries, older adults may face temporary or permanent physical limitations, impacting their independence and mobility. As a result, adjustments to their home environment and daily routine are often necessary. Moreover, the fear of falling again can limit the individual’s participation in social and physical activities, leading to social isolation and a decline in mental health [5,9].

Thus, preventing the consequences of falls is the best way of securing a longer independent life with enhanced physical and emotional well-being for older adult citizens [10,11].

Although physical exercise is one of the most effective methods of preventing falls [10,11], the main barriers to adherence to exercise programs are reported to be a lack of time and inefficient and unsatisfactory exercises [10,12,13]. In this context, the application of exergames has been recommended for improving the physical and cognitive fitness of older adults since they offer a more positive experience with increased motivation in the practice of regular exercise. Moreover, exergaming is claimed to be an enjoyable, interesting, and inexpensive method for improving perceived self-sufficiency and facilitating healthy aging [14,15].

Exergames (combining the words exercise and game) can be defined as digital games that require body movement to be played. They provide an active experience, which functions as a form of physical exercise [16,17,18,19]. Despite the credible benefits of exergaming, there is a substantial gap in our knowledge regarding the efficacy of the intervention method on balance function and on the reduction of fall risk in the older population [20,21].

Currently, exergaming utilizes motion sensors, virtual and augmented reality, and artificial intelligence to create a more immersive exercise experience. Many emerging devices and games allow individuals to access personalized exercise routines, monitor their progress, set goals, and receive real-time feedback [22,23]. With these technological advances, it is possible to better understand variables related to performance in exergames and how they can be used more effectively to improve balance and decrease the risk of falls, in addition to improving the general health, quality of life, and well-being of older adults [23].

For older adults, exergames can provide overall health benefits, improve physical capacity and fitness, and reduce the risk of developing chronic diseases. Additionally, exergames are valuable for motivating physical activity and are important for cognitive stimulation as they can enhance concentration and decision-making. However, exergames pose certain risks for elderly participants, such as injuries, and technological frustrations [22,23,24,25].

Other reviews have been published on the improvements in physical and cognitive function for older adults [20] or on the effectiveness of exergames only in community-dwelling individuals [22,23]. The present systematic review with a meta-analysis is the first to assess and combine different instruments to evaluate the risk of falls and balance and compare the performance of community-dwelling older adults and residents in nursing homes in addition to exploring possible sources of heterogeneity through subgroup analyses.

In this way, the aim of this systematic review was to compile and critically analyze the findings of published studies concerning the efficacy of exergames in improving balance and diminishing the risk of falls in older adults.

## 2. Materials and Methods

### 2.1. Search Strategy

The systematic literature review was based on the Preferred Reporting Items for Systematic Reviews and Meta-Analyses [24] (Appendix A). We had an unpublished search-extraction protocol that did not change over the course of the study. The remit of the search was to include studies that employed exergames as an intervention method to reduce fall risk or the occurrence of falls or for improving the balance of older adults. The expected health-related outcomes were improved balance and/or risk of falls.

### 2.2. Bibliographic Search and Selection of Studies

The searches were performed up to January 2023 using English, Portuguese, and Spanish keyword combinations with Boolean operators (in the Google Scholar, Embase, and PubMed databases) and the PICO approach: population (individuals older than an average of 60 years; residents in community centers or nursing homes), intervention (exergame or electronic gaming platform), comparison (groups with no intervention or interventions with conventional therapies for balance training and/or risk of falls), and outcome (risk score for falls, balance in older adults, and/or the occurrence of falls during the training period) (Table 1). The searches were organized in the databases by the descriptors of each PICO item (Appendix A).

In order to narrow the initial search results, the following filters (automatic tools) were applied to the databases: clinical trials, randomized controlled trials and quasi-experimental studies published in the last 11 years, and full texts (i.e., studies reported only as the abstracts were excluded). Finally, the reference list of the included studies was analyzed to check for further studies to be included. A restriction to studies carried out within the last decade was initiated because of the rapid technological evolution of Exergames. The selection of studies was performed by one researcher (JCL). After the studies were selected, two other researchers (RVF and IMS) analyzed the included studies to ensure proper inclusion. Discrepancies among the researchers who analyzed the studies were resolved by another researcher (VSB).

### 2.3. Inclusion and Exclusion Criteria

The criteria for the inclusion of specified studies were: (i) clinical trials, randomized controlled trials, and quasi-experimental studies evaluating the efficacy of exergames on the fall risk and/or balance in individuals with an average age of 60 years or older who lived in older adult community centers or nursing homes; (ii) interventions with a duration of at least three weeks; (iii) blinded or non-blinded approaches; and (iv) unrestricted publication language. Relevant publications were selected on the basis of the eligibility criteria by reading the titles and abstracts. Following the removal of duplicated studies, the complete texts of the remaining articles were analyzed to select the studies to be included in this review (Figure 1) [24,25].

### 2.4. Data Extraction

Data from full texts of the included articles were extracted by two of the coauthors (JCL and IMS) and verified by all authors. The following information was recorded for each study: year, country, the duration of the experimental phase, population, experimental design, exposure, outcome, the diagnostic method employed in the assessment of balance and fall risk, confounding variables, adherence rate, results, measures of efficacy, discussion, and conclusion.

### 2.5. Data Quality

The Cochrane Collaboration tool was used to analyze the risk of bias from RCT studies [26,27,28,29,30,31]. This tool addresses seven specific domains, as detailed in Figure 2 and Figure 3, related to bias in selection, performance, detection, attrition, and reporting. For each domain, the studies were classified as presenting a low, unclear, or high risk of bias [32]. The risk of bias proportions were calculated as the ratio between the number of domains with a low, unclear, or high risk of bias by the total number of domains of the Cochrane collaboration tool.

For the non-randomized study, the ROBINS-I tool was used [33,34]. The risk of bias was assessed by seven domains: bias due to confounding factors, in the selection of participants for the study, in the classification of interventions, due to deviations from the intended interventions, due to missing data, in the measurement of the outcomes and, in the selection of the reported result.

The small number of studies hindered the assessment of the occurrence of publication bias.

### 2.6. Quality of Evidence

To assess the quality of evidence, the GRADE approach was used [35], by means of the GRADEpro GDT [36]. Evidence was classified as “high”, “moderate”, “low”, and “very low”, depending on five criteria: (i) risk of bias; (ii) inconsistency of effect; (iii) indirect evidence; (iv) inaccuracy; and (v) publication bias. For this analysis, the risk of falls and balance groups were evaluated. Qualitative statements were standardized to describe the different combinations of effect size and certainty of evidence (Appendix A) [37,38].

### 2.7. Data Analysis

The included studies were classified into two groups according to what they assessed: (i) risk of falls (studies that evaluated balance, and other capacities such as lower limb strength, sway speed, etc.) and (ii) balance-only studies.

The studies which used instruments to evaluate the risk of falls (group i) were divided into three categories: (a) the timed up and go (TUG) test, which assesses the normality and velocity of walking in addition to the ability to rise from a chair, ambulate, turn round, return, and sit in the chair again [39,40], (b) the physiological profile assessment (PPA), which involves a series of simple tests of vision, peripheral sensation, muscle force, reaction time, and postural sway and differentiates people who are at risk of falls (“fallers”) from people who are not at risk of falls (“non-fallers”) [41,42], and (c) other methods for the diagnosis of fall risk. The studies that used instruments that assessed balance (group ii) were divided into two categories: (d) Berg balance scale (BBS) studies, which determine static and dynamic balance during a series of predetermined tasks, including rising from a seated position and standing in a bipedal or unipedal stance until balance is lost, with higher scores indicating better balance [43,44], and (e) other instruments that evaluate balance (Appendix A). The choice for these categories was based on the methods that assessed the risk of falls or the most prevalent balance in the studies included in this review.

The adherence rate was calculated as the ratio of the number of participants who completed the intervention and/or control period by the number of participants recruited and/or who started the intervention. (Appendix A).

Meta-analyses were performed separately for (i) the risk of falls and (ii) balance by recording the mean, standard deviation, and sample size values of the intervention and control groups and combining the standardized mean differences (SMD, Hedges’ g—95% confidence intervals) of individual studies in the following subgroups: categories (as previously indicated for risk of falls (“a”, “b”, and “c”) and balance (“d” and “e”)), the execution or not of the intervention in the control group, the place of execution (community centers or nursing homes), and the duration of the intervention. In situations in which the same study used more than one instrument, those which were most frequently used in the literature were kept in the meta-analyses.

Cochran’s Q test and I^2^ statistics were employed to quantify the heterogeneity among the results of the primary studies in the meta-analyses [45]. The Q test was used for the comparison of the effect sizes and the heterogeneity between the subgroups. The studies were combined within the subgroups by means of random effects models. For the execution or not of the intervention in the control group and for the duration of the intervention, there was also a combination of subgroups by means of mixed-effects models [46,47].

All analyses were performed with the aid of Comprehensive Meta-Analysis software version 13 (Biostat, Englewood, NJ, USA).

Since the numbers of studies that evaluated accidental falls among older adults were insufficient for the meta-analysis, the results were narratively discussed by indicating the strengths of associations and their statistical significance.

The materials employed and/or analyzed during the current study are available from the corresponding author upon reasonable request.

**Figure 3 healthcare-11-01872-f003:**
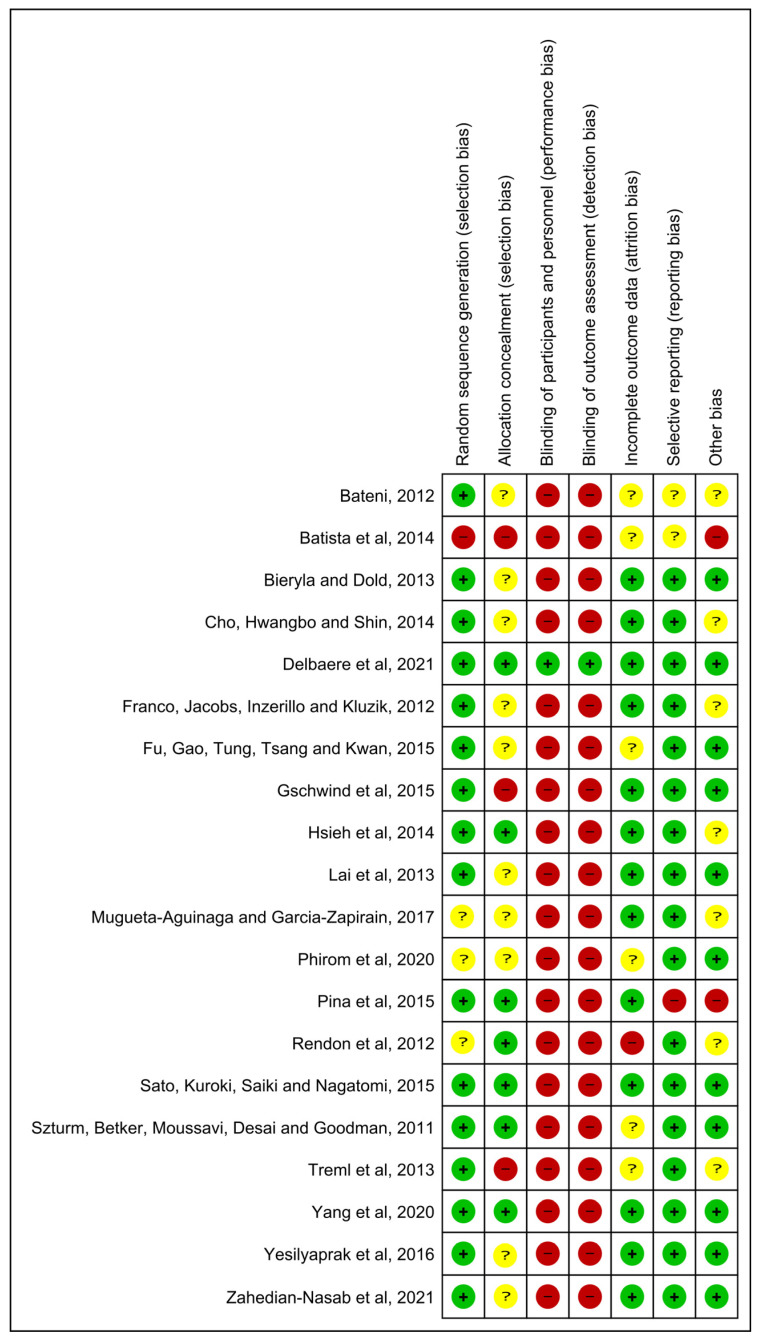
Summary of the analysis of the studies included in the review with respect to the risk of bias: low risk (+), unclear risk (?), and high risk (-) [12,13,16,17,19,42,48,49,50,51,52,53,54,55,56,57,58,59,60,61].

## 3. Results

### 3.1. Studies Included

The bibliographic search resulted in 20 articles being selected for inclusion in the review (Figure 1) [25]. Studies conducted in 12 countries were included: four from the United States [12,48,49,50], three from Brazil [51,52,53], three from Taiwan [16,54,55], two from Australia [19,56], and one each from Hong Kong [57], Canada [58], South Korea [59], Japan [17], Spain [13], Turkey [60], Iran [61], and Thailand [42]. Six studies [12,13,50,59,62,63] were conducted in nursing homes. The remaining 14 studies were conducted with community-dwelling older adults (Appendix A).

### 3.2. Quality of Evidence and Risk of Bias

Considering the risk of bias within the studies, it was observed that 49.62% of the information extracted presented a low risk, 19.55% presented an unclear risk, and 30.83% presented a high risk (Figure 2 and Figure 3). Only one article had a quasi-experimental design [51]. Four domains of ROBINS-I indicated a low risk of bias, and three domains indicated a moderate risk of bias (Appendix A).

Regarding the quality of evidence evaluated by the mean of the GRADE, low quality was observed for the risk of falls and for balance (Appendix A).

### 3.3. Participants

The total sample population comprised 1303 participants, with a median number of 65 per study. One study [62] had an intervention of 12 months, with an additional 12-month follow-up. The interventions in the other publications varied in duration, ranging from 3 to 16 weeks (with a mean of 7.15 weeks), occurring 2 to 3 times per week, and consisting of 9 to 36 sessions (with a mean of 18.57 sessions). Specifically, three studies [12,13,51] had interventions lasting 3 weeks, one study lasted 4 weeks [50], three studies lasted 5 weeks [53,55,57], five studies lasted 6 weeks [52,56,59,62,63], three studies lasted 8 weeks [54,60,61], one study lasted 9 weeks [19], three studies lasted 12 weeks [16,17,44], and one study lasted 16 weeks [58]. Ten studies implemented the interventions twice a week [12,13,17,19,51,53,55,57,60,63], while eight studies conducted the interventions three times a week [16,44,50,52,54,59,61,62]. One study applied the interventions five times a week [55], and one study did not provide information regarding the weekly frequency [58]. A total of 1117 participants completed the training period or returned for the final assessment. In general, the adherence rate was 85.72% (Appendix A).

The ages ranged between 60.25 [54] and 85.47 years [13] in the experimental groups and between 67.2 [52] and 83.11 years in the control groups [13]. Rendon et al. (2012) [50] did not present the mean age of the participants but reported that their ages ranged between 60 and 95 years, and Batista et al. (2014) [51] studied participants with a mean age of 68.1 years. All studies involved both male and female participants except for that reported by Batista et al. (2014) [51], in which only women took part.

### 3.4. Interventions

The Nintendo Wii Fit Balance Board was the controller most frequently used in training programs (nine studies) [12,48,49,50,51,52,53,57,59]. The description of the controllers used in each study can be found in Appendix A under “Used Technology”. In seven studies, the control groups received no training [12,13,17,51,52,56,61], while the comparison group received some interventions, such as conventional balance training [62], conventional proprioception training [55], stationary bicycle exercises [54], aerobic exercises [54], usual care and/or routine activities [19,63], and interventions with educational material [44,58]. In six studies, the older adults received conventional training [16,50,57,59,60,62]. The mean duration of the interventions that assessed the risk of falls was 9.6 weeks, with a range between 3 and 52 weeks. The mean duration of the interventions that assessed balance was 9.2 weeks, with the same variation in the risk of falls.

To assess the risk of falls, TUG was the tool most frequently used, in 11 studies [12,16,19,42,52,54,55,58,60,61,63]. Another four studies used the PPA to assess the risk of falls [19,42,57,63]. In addition to TUG and the PPA, 11 other instruments were used to assess the risk of falls. To assess balance, the BBS was the most used instrument, appearing in 12 studies [12,16,17,48,49,51,52,53,54,58,60,61]. A further 10 different instruments were used to assess balance. In four instruments, balance was assessed with the participants with their eyes closed and eyes open. Of the included studies, 17 [12,13,16,17,19,41,47,48,49,51,54,55,56,57,58,59,60] were randomized clinical trials (RCTs). One study was a clinical trial (CT) [53] and two studies had a quasi-experimental design [50,52].

### 3.5. Comparison

In seven studies, the control groups did not practice any other exercise [12,13,17,51,52,56,61]. In the other studies, the comparison groups practiced other exercises; most of them had traditional training for the risk of falls and/or balance.

### 3.6. Outcomes

For the risk of falls, a significant improvement in exergame groups was observed in relation to the control groups for TUG (Hedges’ g = 0.217, CI = 0.023; 0.411, *p* = 0.028), PPA (Hedges’ g = 0.382, CI = 0.004; 0.760, *p* = 0.048), and the other instruments (Hedges’ g = 0.216, CI = 0.040; 0.392, *p* = 0.016) (Figure 4). In the risk of falls outcome, significant heterogeneity was observed only in the PPA subgroup (I^2^ = 76.006%; *p* = 0.022). In the TUG groups (I^2^ = 25.71%; *p* = 0.191) and the other instruments (I^2^ = 13.241%; *p* = 0.318), no significant heterogeneity was observed. (Appendix A).

For balance, a significant improvement in exergame groups was observed in relation to the control groups for the BBS (Hedges’ g = 0.317, CI = 0.087; 0.546, *p* = 0.007) and the other instruments (Hedges’ g = 0.329, CI = 0.186; 0.472, *p* < 0.000) (Figure 5). In the balance outcome, significant heterogeneity was observed in both subgroups: the BBS (I^2^ = 61.892%; *p* = 0.034) and the other instruments (I^2^ = 74.055%, *p* = 0.0675).

Analyses of the other subgroups are presented in Appendix A. In general, except for the subgroup that separated the studies on the risk of falls by place of execution (nursing homes or community centers), the adopted subgroups were not able to explain the heterogeneities in the individual studies. Significant heterogeneities remain within groups, which does not occur between groups. Although there were differences in the presence or absence of statistical significance, all subgroups, both for the risk of falls and for balance, showed positive results with the use of exergames. Only two studies [19,57] analyzed accidental falls among older adults. Fu et al. (2015) [57] reported that the incidence of falls was significantly lower in the exergame intervention group compared with the control group trained with conventional exercises (0.54 vs. 1.52 falls per person years, respectively). Delbaere et al. (2021) [19] reported that the risk of falls in the first 12 months of follow-up was lower (incidence rate ratio = 0.82) in their intervention group, although without statistical significance. Considering the 24-month follow-up, the difference (incidence rate ratio = 0.84) became statistically significant.

## 4. Discussion

The aim of this systematic review was to compile and critically analyze the findings of published studies concerning the efficacy of exergames in improving balance and reducing the risk of falls in older adults. The types of games, the outcome assessment tools, and the duration of intervention differed greatly between the studies, but the overall and subgroup results suggest that exergames have positive efficacy on balance and reduction of fall risks in older adults.

According to the meta-analysis, exergames were able to promote statistically significant positive effects on balance and the risk of falls in the groups submitted to intervention. Previous systematic reviews about the efficacy of exergames have already shown similar results [22,64,65]. The present review showed that the improvement occurs regardless of the instruments adopted to assess the results, the way in which the intervention was performed in the control groups, and the execution time in the short-term studies. However, care is needed with the interpretation of the results due to the low quality of the evidence. Despite this, it is important to note that only the study of Delbaere et al. (2021) [19] followed participants over a long period (24 months). In this study, the program significantly reduced the rate of falls after two years of follow-up. On the other side, meta-analysis of the present review showed that the reduction in the risk of falls was not significant after 12 weeks of follow-up. In this way, the results are not conclusive but indicate that exergames can be less effective in interventions designed to last for prolonged periods and this question must be better investigated.

Exergames were associated with an improvement in outcome measures that are used to predict the risk of falls. One study in particular showed that that the incidence of falls was lower in participants who trained with exergames [57]. Such an achievement may be explained by the visual and auditory stimuli provided by the games, which favor feedback mechanisms in the learning process and the retention of motor tasks performed in real-life situations [66]. The simultaneous feedback provided by exergames allows players to correct and synchronize their movement in real-time and, consequently, encourages better exercise performance and rehabilitation. In addition, the delivery and control of a dynamic stimulus is promoted by the exergaming environment [57]. Hence, the effectiveness of exergames, especially with respect to balance, can be explained by their efficiency and unpredictability. Another important aspect is that exergames seem to reduce the fear of falling in older individuals [61]. One of the advantages of exergames lies in the possibility of combining the specific tasks of the games used (sky slalom, sky jump, and soccer heading) with coaching instructions, such as bending forward, changing direction, and shifting weight. As a result, besides achieving balance control during the exercises, exergames allow participants to attain more effective movement or posture [20,67,68]. Older adults, due to the inherent characteristics of aging, experience reductions in muscle strength and balance. A lack of physical activity can exacerbate these losses, increasing the risk of falls. Therefore, exercise programs that involve balance training and muscle strengthening in sedentary older adults are expected to reduce the risk of falls [57].

One reason for the improvement observed in the groups that engaged in exergames is the greater appeal of interactive video games to older adults. This appeal contributes to increased performance in functional activities, which have been shown to effectively improve dynamic balance [57]. Other studies have also indicated that motivation and interest result in greater awareness and control of balance [69,70].

The duration of the intervention can play a crucial role in determining the effectiveness of exergaming. In this review, interventions lasting up to 12 weeks significantly reduced the risk of falls. However, interventions lasting 12 weeks or more were less common in the literature and did not show significant differences in fall risk reduction. This highlights the importance of considering longer interventions in future studies. A lack of physical activity can result in the weakening of the lower limbs and compromise balance. Thus, any intervention that includes physical activity can offer benefits for balance and reduce the risk of falls [55]. Our results show that the subjects taking part in exergaming had better results than those in the control groups with other exercises. However, the high heterogeneity, together with the low quality of the evidence, does not make it possible to say that exergames are superior to other practices. The publication of more studies will be useful for better clarification of this question.

Despite the benefits of exergaming, it is important to adopt safety measures and to always monitor participants, especially at the start of training, because the gaming platform and the screen may cause dizziness and disorientation which can cause accidents [66]. Moreover, as pointed out by Vojciechowski et al. (2018) [21], prescriptions of exercises involving exergaming typically lack specifications of, for example, the type of game, the number of sets and repetitions, and the duration, frequency, and intensity of training. Indeed, the studies included in this review varied considerably with respect to many of these parameters in addition to the outcomes analyzed, but none of the investigations examined training intensity.

Older adults in nursing homes often experience a rapid decline in daily functioning, leading to cognitive impairment, depression, and sleep disorders [71]. Despite the known benefits of physical activity and social interaction, these individuals spend a significant amount of time alone in their rooms without engaging in activities. This lack of physical activity and daily routine contributes to a significant functional deficit [71,72,73,74,75,76,77]. In our meta-analysis, the location where the interventions were conducted was the only subgroup that adequately explained the differences in the study results for fall risks. Minor variations were observed within the results of studies conducted in community centers or nursing homes, but significant variations were observed between these groups. This indicates that the specific characteristics of older adults living in each location contribute to differences in the effects of exergames.

Exergames have been found to be effective in significantly reducing fall risk and improving balance among older adults who live in community centers. However, there were no statistically significant effects on balance among individuals in the nursing homes group. This could be attributed to the lower statistical power of the analysis with studies conducted in nursing homes, which were less common in the literature. Therefore, although the findings suggest that exergames are important tools for improving balance, additional studies are still necessary.

Several reviews have examined the effectiveness of exercise in reducing the risk of falls and improving balance in older adults [22,38,39,78,79,80], all of which reported positive outcomes. However, only a couple of reviews [22,39] specifically discussed the use of exergames without comparing the issues analyzed in our review, such as the tools, interventions, duration of the interventions, and place of execution (community centers and nursing homes). Overall, the literature demonstrates that exercise is an important tool for reducing falls and improving balance in older adults, but it remains unclear whether exergames are superior to other exercise interventions [22,39]. Despite this, the published studies and the present review show that exergames are considered acceptable for community-dwelling older adults.

The assessment tools used in the studies evaluated a combination of balance, sway velocity, and lower limb strength, but these factors may not be sufficient to predict fall risk in older adult citizens [11]. Furthermore, some criticism may be justified with regard to the lack of blinding in the studies analyzed. Although it would be exceedingly difficult, if not impossible, to conceal the identities of the participants from the researchers who applied the tests, blinding could have been incorporated into the study design by masking the identities of the assessors from the individuals who performed the statistical analyses. In addition, many of the studies (~75%) employed small sample populations (≤40 participants) or were classified as having a high risk of bias (32.85%). The studies presented limited data on the socioeconomic characteristics, lifestyle, or health of the participants, while none of them analyzed the effectiveness of the exergame intervention according to age group. Another important limitation was the non-execution of publication bias analysis due to the small number of included studies.

In light of the above, future randomized clinical trials should encompass some fundamental protocols including: (i) increasing the sample size with blinding and division of participants into control, conventional training, and exergaming groups; (ii) standardizing the exergames by separating “commercial games” from specific health-related games aimed at training for balance and/or risk of falls; (iii) monitoring exercise intensity in all training groups; (iv) performing long-term interventions; (v) following up participants after the intervention to verify the retention of learning and physical gains obtained during the execution of the exergames and the ability to apply such information in daily life; (vi) analyzing variables such as prescribed medication, housing conditions, daily living habits, history of illnesses, and type of work which may impact on the intervention results; (vii) assessing the incidence of falls over time; (viii) monitoring participants according to their age group; (ix) performing better quality studies, using methods for blinding the participants and researchers; and (x) exploring various types of exergames (games and consoles) and examining other factors that may promote more effective effects of exergame use.

## 5. Conclusions

Although there are many gaps in our knowledge about the benefits of exergaming, especially concerning long-term interventions and the effectiveness of exergames compared to other forms of exercise for older adults, we conclude that there is a low certainty of evidence that exergames are suitable for improving balance and reducing fall risk in older adults. Further studies on the use of exergames should be pursued and their use as a collective health strategy should be evaluated, especially in older adult recreational centers, nursing homes, and basic health units.

## Figures and Tables

**Figure 1 healthcare-11-01872-f001:**
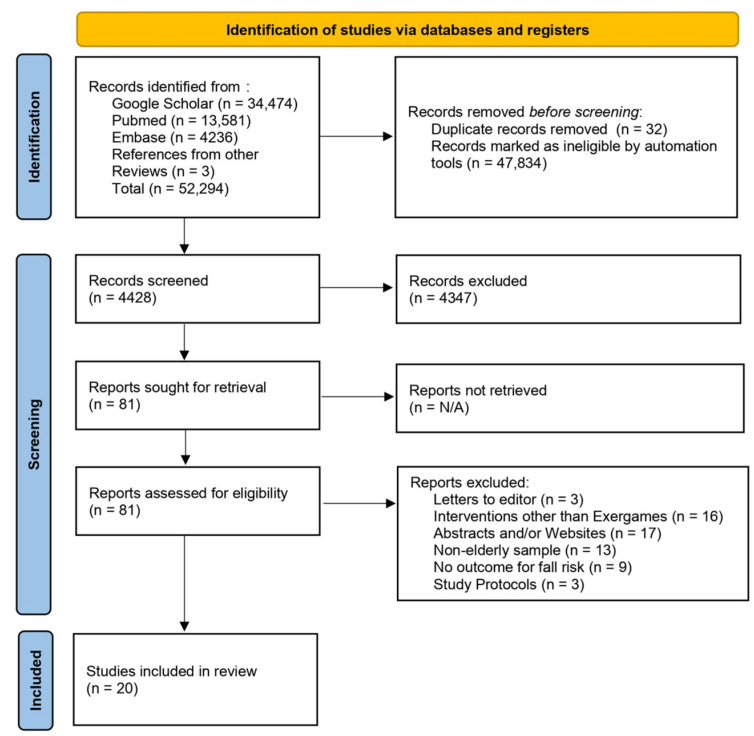
PRISMA 2020 flow chart illustrating the systematic review process [24,25].

**Figure 2 healthcare-11-01872-f002:**
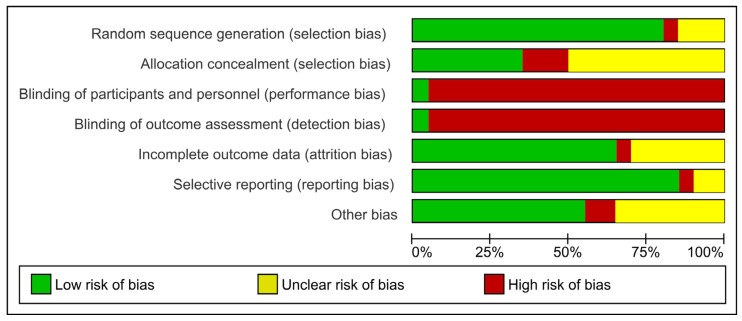
Quality of studies and risk of bias according to the Cochrane Collaboration tool.

**Figure 4 healthcare-11-01872-f004:**
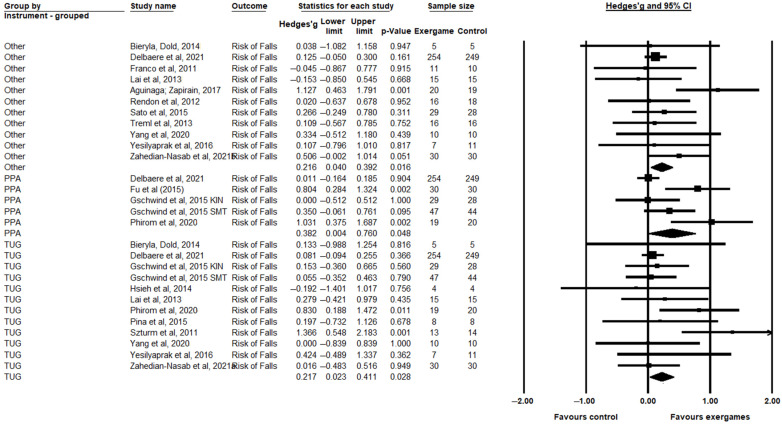
Comparison between the intervention (exergaming) and control groups (conventional training or without intervention) evaluated regarding the risk of falls, in the time up and go (TUG), physiological profile assessment (PPA), and other categories, showing differences with standardized means (SMD) and 95% confidence intervals (CI) [12,13,16,17,19,42,49,50,52,53,54,55,56,57,58,60,61].

**Figure 5 healthcare-11-01872-f005:**
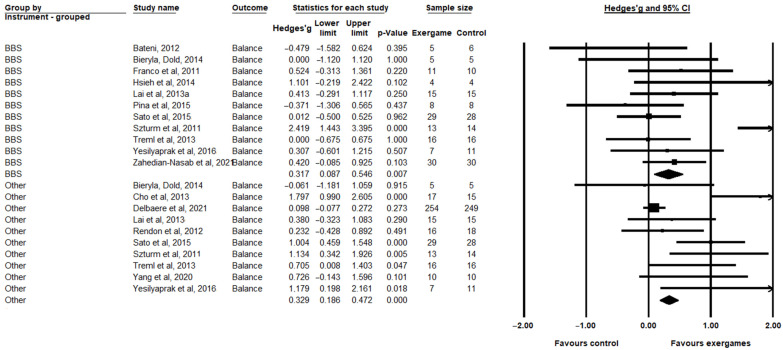
Comparison between the intervention (exergaming) and control groups (conventional training or without intervention) evaluated regarding balance, in the Berg balance scale (BBS) categories and the other categories, showing the standardized mean differences (SMD) and confidence intervals (CI) of 95% [12,16,17,19,48,49,50,52,53,54,55,58,59,60,61].

**Table 1 healthcare-11-01872-t001:** Keywords used in the literature search.

PICO Protocol	Keywords
Population	*MeSH terms:* “Aged” OR “Frail Elderly” *Entry terms:* (Elderly) OR (Elderly, Frail) OR (Frail Elders) OR (Elder, Frail) OR (Frail Older Adults) OR (Adult, Frail Older) (Frail Older Adult) OR (Oldest Old)
Intervention	*MeSH term:* “Video Games” OR Exergaming *Entry terms:* (Game, Video) OR (Games, Video) OR (Video Game) OR (Computer Games) OR (Computer Game) OR (Game, Computer) OR (Games, Computer) (Exergame) OR (Exergames) OR (Exergamings) OR (Active-Video Game) OR (Active Video Gaming) OR (Active-Video Gamings) OR (Gaming, Active-Video) OR (Virtual Reality Exercise) OR (Exercise, Virtual Reality) OR (Virtual Reality Exercises)
Comparison	Any group
Outcome	*MeSH term:* “Accidental Falls” OR “Postural Balance” *Entry terms:* (Falls) OR (Fall) OR (Falling) OR (Falls, Accidental) OR (Accidental Fall) OR (Fall, Accidental) OR (Slip and Fall) OR (Fall and Slip) OR (Balance) OR (Posture Equilibrium) OR (Equilibrium, Posture) OR (Posture Equilibriums) OR (Balance, Postural) OR (Postural Equilibrium) OR (Equilibrium)

*MeSH—medical subject headings*.

## Data Availability

The data presented in this study are available in the Appendix A of this study.

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
