# Peer review of "Exergames in Older Adult Community Centers and Nursing Homes to Improve Balance and Minimize the Risk of Falls in Older Adults: A Systematic Review and Meta-Analysis"

_healthcare, 2023, doi:10.3390/healthcare11131872_

Round 1

Reviewer 1 Report

The systematic review and meta-analysis aim to describing the efficacy of exergames to improve balance or reduce the risk of falls in individuals above 60 years of age, residents in community centers or nursing homes. Based on 20 included studies they conclude that there is a low certainty of evidence that exergames are suitable for improving balance and reducing fall risk in older adults.

All in all I think this is a well written review that adds value to the field of exergaming in older adults. The story is well told, and the methods are solid and well performed with a good search strategy and quality assessments. However, I have one major comment to this review with regards to the presentation of the results. In the title, and the aim, the authors distinguish between residents in community centers and nursing homes. However, there is no mention of which studies includes with residents in the results. In my experience, both age and level of physical and cognitive functioning might be quite different between residents living in community homes (or is it even community-dwelling older adults attending community centers just during daytime?), and those living in a nursing home. I think the review would benefit, and be even stronger, if the authors present the differences between these two living facilities as well in their results. It is presented in the characteristics of the studies in some of the studies in Supplementary material 4, but its not described for each study here either. Is there a difference in the included participants, the type of intervention used, and not at least in the outcome measured used? Older adults attending community centers might actually have such good physical function that they reach a celling effect on some of the balance tests used.

Interestingly, the part of community centers and nursing homes seems to also disappear a bit in the discussion. I do think this is the part that might distinguish the current review from previous reviews and hence, should be more visible also in the discussion. As it is now, the discussion is a bit weak, and does not really reflect on the job that has been done to conduct the review. I would highly recommend the authors to go through the discussion and rewrite it, adding points like for instance the differences between care level, how different games seem to affect differently, what would be clinically valuable to use in this group based on the current results, and is there really a difference between those games used for balance and those used for minimizing fall risk? Isn’t that two sides of the same matter? By strengthening your discussion, you also have the possibility to be a bit more direct in your conclusion. Stating that there are gaps remaining in our knowledge is a bit vague.

I really like the idea of this review, and I think the job is well done so I will highly recommend putting in the effort to the results and the discussion to lift these parts up to the standard it deserves.

Minor comments:

-          In the PRISMA flow chart in figure 1 you write in the first box “registers (n=52 294)”. Based on the number this is the total, but using registers is a bit confusing. Just write Total.

-          Also in Figure 1 you state “records marked as ineligible by automatic tools” – which automatic tools? This is not explained in the methods, please add.

-          I think you should consider reporting duration of sessions and number of sessions per week in the results. This will give the reader a better understanding of the amount of exercise performed and hence, how valid the results are. In your discussion you have a section about this (line 351-356) which would benefit from you actually showing these results in your paper. Also, it would give better grounds for a more through discussion around the matter.

-          The use of references is a bit varying in the results section. Please add references in section where you mention things that have been used by different studies such as in 3.4 “…while, the comparison group received some intervention, such as conventional balance training (ref), conventional proprioception training (ref), stationary bicycle (ref), aerobic exercises (ref), usual care and/or routine activities (ref) and interventions with educational material for the risk of falls (ref).”

The language is overall fine, but there are a few grammatical errors and some prepositions lacking so I will recommend you to read through the paper thoroughly.

Reviewer 2 Report

REVIEWER Comments:

The authors aim to critically analyze the role of exergames in improving balance and reducing fall risk among older adults. The research question is relevant and timely, and the methodology adopted to answer this question is robust. However, there are several areas where the manuscript could benefit from revisions and further clarification.

Introduction

1.      The explanation of exergames and their current technological advancements is well-done. However, consider discussing the potential benefits or drawbacks of these technologies in the context of older adults.

2.      The initial paragraphs effectively lay the foundation by discussing the growing proportion of older adults and the associated challenges. Your discussion about accidental falls, their causes, and impacts is well-presented. However, consider diving a bit deeper into the ramifications of these falls on health services and the family structure. This will further stress the urgency of your study.

Methods

1.      The authors could elaborate on the process of data extraction, particularly concerning how discrepancies between reviewers were resolved.

Discussion:

1.      The discussion is comprehensive and presents the findings of the systematic review well. However, here are some specific comments and suggestions to enhance the section:

2.      The explanation of the findings in relation to the study's aim is done well. The discussion on the impact of exergames on balance and the risk of falls in older adults is clear. However, consider expanding on the factors that contributed to the variation in results across the studies. For instance, how did the type of exergame or the differences in the intervention's duration affect the outcomes?

3.      The comparison of your findings to those of previous systematic reviews is important. While the discussion is good, consider providing more details about how your findings either corroborate, extend, or contradict previous work.

Conclusion

1.      The authors recommend future studies on the use of exergames and their implementation as a collective health strategy. While this is a pertinent suggestion, it might be beneficial to specify what particular aspects need to be explored in future research. For instance, do we need more randomized controlled trials, long-term follow-up studies, or studies investigating the specific types of exergames that are most beneficial?

Round 2

Reviewer 1 Report

Thank you for your extensive work on the resubmitted article. It reads much better and is much more cohesive. Great work.

There are some minor errors in the language that might benefit from having a native English-speaking person read through it before publication.